# Rotator Cuff Muscle Imbalance in Patients with Chronic Anterior Shoulder Instability

**DOI:** 10.3390/diagnostics14060648

**Published:** 2024-03-19

**Authors:** Du-Han Kim, Ji-Hoon Kim, Chul-Hyun Cho

**Affiliations:** Department of Orthopedic Surgery, Keimyung University Dongsan Hospital, Keimyung University School of Medicine, Daegu 42601, Republic of Korea; osmdkdh@gmail.com (D.-H.K.); dortn3217@naver.com (J.-H.K.)

**Keywords:** shoulder instability, rotator cuff, muscle imbalance, dislocation

## Abstract

(1) Background: Both intra-articular pathologies and muscle imbalance can be a cause of shoulder instability. The purpose of this study is to examine the cross-sectional areas of the rotator cuff muscle in patients with acute and chronic anterior shoulder instability and to determine the associations between imbalance and chronicity of the rotator cuff. (2) Methods: Patients with confirmed dislocation of the anterior shoulder were included. The patients were divided into two groups according to the time between the initial dislocation event and when MRI imaging was performed Measurements of the rotator cuff muscle areas were performed in the scapular Y view and glenoid face view using MRI. (3) Results: A total of 56 patients were enrolled. In the Y view, a larger area of supraspinatus muscle was observed in the chronic group compared with the acute group (17.2 ± 2.3% vs. 15.6 ± 2.2%, *p* = 0.006). However, a smaller area of subscapularis muscle was observed in the chronic group (47.1 ± 3.5% vs. 49.6 ± 5.3%, *p* = 0.044). Using the glenoid face view, a larger area of supraspinatus muscle was observed in the chronic group than in the acute group (18.5 ± 2.5% vs. 15.8 ± 2.2%, *p* < 0.001). However, a smaller area of subscapularis muscle was observed in the chronic group (41.6 ± 3.2% vs. 45.6 ± 4.4%, *p* < 0.001). (4) Conclusion: Larger areas of supraspinatus muscle compared with acute instability were observed in patients with chronic anterior shoulder instability. In contrast, a smaller area of subscapularis muscle was observed in the chronic group.

## 1. Introduction

The glenohumeral joint, one of the most unstable joints in the body, can be susceptible to various types of injury, particularly dislocation [1,2,3,4]. Secondary injury of the articular labrum, cartilage, capsule, and glenohumeral ligaments can result in shoulder instability [3,5]. In particular, secondary intra-articular injuries are more commonly reported in patients with chronic shoulder instability [3,6]. Deficiencies in static and/or dynamic stabilizers of the humeral head on the glenoid can be a cause of shoulder instability [7]. In addition, an unstable joint may present with both intra-articular pathologies as well as muscle imbalance. This type of imbalance is known as arthrogenic muscle inhibition [8]. Mechanisms for this type of inhibition include alteration in muscle resting motor thresholds, changes in the discharge of articular sensory receptors, altered excitability of the spinal reflex, and abnormal cortical activity [9,10,11].

One example of this muscle inhibition can be observed in the knee joint, particularly in the case of failed activation of the quadriceps after rupture of the anterior cruciate ligament [8]. Several authors have reported an association between hamstring overactivity and dyskinesia with weakness of the quadriceps in cases of arthrogenic muscle inhibition [12,13]. This has been attributed to the flexion reflex spinal pathway, which causes increased activity in flexor muscles and reduced activity in extensor muscles. One study reported an association of greater hamstring coactivation with significantly worse knee function [13]. Another study reported an association of hamstring fatigue exercises in patients with arthrogenic muscle inhibition following reconstruction of the anterior cruciate ligament with a significant increase in quadriceps strength [14].

The rotator cuff muscles exert compression force in the glenohumeral joint, counteracting the superior shear force applied by the deltoid [15]. In addition, the force balance between the subscapularis and infraspinatus and teres minor, known as the transverse force couple, provides pivotal support to the anteroposterior stability of the glenohumeral joint [16,17]. Numerous studies have reported on the relationship between shoulder instability and changes in functional muscle. In particular, the conduct of research on the balance of the rotator cuff force and measurement of isokinetic power, using various methods is currently ongoing [18,19,20,21]. However, the results are still unclear and inconsistent, which could be due to the limitations of the isokinetic power test due to compensation from other muscles such as the deltoid or pectoralis muscle, for weakness in one rotator cuff muscle. Therefore, the use of a method for direct measurement that involves isolation of the rotator cuffs has been attempted. A recent study using magnetic resonance imaging (MRI) examined cross-sectional areas of the rotator cuff and deltoid muscle in three types of shoulder instability (anterior, posterior, and multi-direction) [22]. However, no study on analysis of changes in the rotator cuff according to the chronicity of shoulder anterior instability has been reported.

The purpose of this study was to examine the cross-sectional areas of the rotator cuff muscle in patients with acute and chronic anterior shoulder instability and to determine the associations between imbalance and chronicity of the rotator cuff. According to our hypothesis, patients with chronic anterior shoulder instability would present with a larger area of infraspinatus (ISP) and teres minor (TM) muscle relative to the subscapularis (SSC).

## 2. Materials and Methods

### 2.1. Patient Selection

This study was approved by the institutional review board of our hospital (IRB No: 2022-09-053). All patients provided written informed consent.

A retrospective review of the medical records of patients with anterior shoulder instability between December 2014 and September 2021 was conducted. The diagnosis of anterior shoulder instability was based on the patient’s medical history, simple radiographs, and MRI. Patients who met the following criteria were included in the study: (1) 18–50 years, and (2) patients with confirmed dislocation of the anterior shoulder. Exclusion criteria included patients who: (1) had no preoperative MRI or a poor-quality image that was difficult to measure, (2) had concomitant fractures, (3) had acute or chronic rotator cuff tears, and (4) had a previous history of shoulder surgery or trauma.

Participants in the study were categorized according to two groups, based on the duration between the initial dislocation event and when MRI imaging was performed. Specifically, patients who underwent MRI imaging within a short timeframe (defined as 1 to 30 days post-dislocation) were classified as the ‘acute group’, while those who underwent MRI after a much longer interval, extending over a period of two years since the first dislocation event, were classified as the ‘chronic group’.

### 2.2. Muscle Area Measurement

MRIs of patients’ shoulders were obtained from a consecutive cohort of scans acquired using a 1.5 T MRI system (Siemens Magnetom Avanto System; Siemens Medical, Erlangen, Germany). Close cooperation with the department of radiology was maintained during the conduct of this study to ensure precise measurements; oblique-sagittal T2 TSE MRI scans were obtained perpendicular to the long axis of the scapular body; the slice thickness of images was 3.6 mm. Image parameters included a repetition time of 569 ms, an echo time of 17 ms, and a 180 × 180-voxel field of view. The contours of the supraspinatus (SSP), SSC, and ISP/TM muscles were traced by hand using PACS software (Picture Archiving and Communication System, DICOM version 3.0 INFINITT, Infinitt Healthcare, Seoul, Republic of Korea). Because the borders of the ISP and TM were often indiscernible, these muscles were combined as a single muscle unit [23,24].

The borders of three muscle groups were traced on the Y-shaped scapular view in the parasagittal plane (Figure 1). Calculation of the results for each muscle was then performed automatically by the software. Tracing of the three muscle contours in the glenoid face view, with a clearer view of the face of the glenoid was performed (Figure 2). After performing measurements in three muscle groups, the percentage of each muscle was calculated with respect to the total rotator cuff muscle.

Measurements of images were performed independently by two experienced orthopedic surgeons (DHK and JHK). A randomized analysis for the evaluation of interobserver reliability was repeated three months later.

### 2.3. Statistical Analysis

Statistical analysis was performed using SPSS 24.0 software (Version 23.0; IBM Co., Armonk, NY, USA). Two methods were used to assess the reliability of the measurements. First, the test–retest reliability was evaluated, followed by an assessment of internal consistency reliability, according to the method outlined by Bland and Altman for the calculation of inter- and intraclass correlation (ICC) [25]. Levene’s test was also applied to determine whether continuous variables showed equal variances. An independent *t*-test was performed for data that showed a normal distribution and involved paired observations, which allowed for a rigorous comparison of differences between groups. In addition, in comparison to categorical variables, the Chi-squared test was used to determine statistically significant associations or differences between groups. The threshold for determining statistical significance was set at *p* < 0.05, meaning that results with *p*-values below this threshold were considered statistically significant, while those above it were not. The use of this comprehensive statistical approach ensured the robustness and reliability of the findings.

## 3. Results

A total of 56 patients (28 patients in the acute group and 28 patients in the chronic group) finally met the criteria for inclusion in the study. No differences in demographic variables were observed between the two groups, except for the number of dislocation events. Patients in the chronic group reported between three to thirty episodes of dislocation (mean, 8.8 ± 7.3) (Table 1).

### 3.1. Inter-Observer and Intra-Observer Reliability

The inter-observer reliability of structures was as follows: (1) Y view SSP (ICC = 0.957, *p* = 0.0001), (2) Y view SSC (ICC = 0.929, *p* = 0.0001), (3) Y view ISP/TM (ICC = 0.914, *p* = 0.0001), (4) glenoid face view SSP (ICC = 0.959, *p* = 0.0001), (5) glenoid face view SSC (ICC = 0.901, *p* = 0.0001), and (6) glenoid face view ISP/TM (ICC = 0.898, *p* = 0.0001). And the intra-observer reliability of structures was as follows: (1) Y view SSP (ICC = 0.881, *p* = 0.0001), (2) Y view SSC (ICC = 0.853, *p* = 0.0001), (3) Y view ISP/TM (ICC = 0.869, *p* = 0.0001), (4) glenoid face view SSP (ICC = 0.911, *p* = 0.0001), (5) glenoid face view SSC (ICC = 0.899, *p* = 0.0001), and (6) glenoid face view ISP/TM (ICC = 0.904, *p* = 0.0001) (Table 2).

### 3.2. Y View Analysis

A larger area of SSP muscle was observed in the chronic group compared with the acute group (17.2 ± 2.3% vs. 15.6 ± 2.2%, *p* = 0.006); however, a smaller area of SSC muscle was observed in the chronic group (47.1 ± 3.5% vs. 49.6 ± 5.3%, *p* = 0.044). No statistical significance was observed in the ISP/TM muscle group.

### 3.3. Glenoid Face View Analysis

A larger area of SSP muscle was observed in the chronic group compared with the acute group (18.5 ± 2.5% vs. 15.8 ± 2.2%, *p* < 0.001); however, a smaller area of SSC muscle was observed in the chronic group (41.6 ± 3.2% vs. 45.6 ± 4.4%, *p* < 0.001). No statistical significance was observed in the ISP/TM muscle group (Table 3).

## 4. Discussion

According to the findings of the current study, a larger area of SSP muscle was observed in patients with chronic anterior shoulder instability compared to those with acute shoulder instability. In contrast, a smaller area of SSC muscle was observed in the chronic group.

Any disruption of the static restraints such as the glenoid labrum or glenohumeral ligaments can compromise the synergistic relationship that results in glenohumeral stability, causing overloading of dynamic stabilizers [26]. In particular, excessive translations of the humeral head at the glenoid can be prevented by rotator cuff muscles, which are critical to the maintenance of dynamic stability. Therefore, an analysis of the relationship between rotator cuff function and shoulder instability was performed by several authors.

Saccol et al. reported on a study comparing rotator cuff muscle strength in athletes with anterior shoulder instability to that of healthy, age- and sport-matched athletes. According to their findings, athletes suffering from anterior shoulder instability exhibited weaker internal and external rotator power compared to healthy athletes in the control group. They concluded that their results could be attributed to anterior shoulder instability [21]. These changes might persist even after surgery. In a study of rotator cuff muscles, Rhee et al. focused on the external rotation power after surgery [20]. A total of 104 patients who underwent arthroscopic surgery were included in their study. The results of measuring the internal and external rotation strength in patients who underwent arthroscopic capsulolabral reconstruction showed that external rotation power remained weak one year after surgery [20]. Their findings demonstrated the importance of external rotation and internal rotation strengthening exercises with greater emphasis on exercises for external rotation to prevent positive apprehension after capsulolabral reconstruction [20]. Amako et al., who conducted a study to determine the period of time required for recovery of rotator cuff muscle strength after arthroscopic Bankart repair [27], evaluated isokinetic concentric shoulder rotational muscle strength in 50 patients diagnosed with recurrent dislocations of the glenohumeral joint who underwent treatment with arthroscopic Bankart repair. They reported that isokinetic strength after surgery had recovered to preoperative levels by six months for external rotation and 4.5 months for internal rotation [27]. Lee et al. reported on a comparative study on patients with and without arthroscopic Bankart repair [19]. A total of 32 patients with shoulder instability who underwent arthroscopic Bankart repair were compared with 32 asymptomatic volunteers using an isokinetic device at an angular velocity. According to their conclusion, full recovery of neuromuscular control of internal and external rotations had not been achieved one year after surgery in patients who underwent Bankart repair [19]. A different result was reported in a similar study conducted by Tahta et al. [28], who analyzed 56 shoulders with Bankart lesions that were operated using arthroscopic techniques. Assessment of muscle activity for four distinct muscles (SSP, ISP, TM, and deltoid) was performed using surface EMG electrodes. They concluded that only internal rotation strength was reduced at 24 months after arthroscopic Bankart repair [28]. Edouard et al. reported on an analysis of the association between isokinetic muscle strength and glenohumeral joint instability in patients with recurrent (chronic) anterior instability who did not undergo previous surgical treatment [29]. Their study included 37 patients with unilateral recurrent anterior post-traumatic shoulder dislocation and 11 healthy non-athletic subjects. As per their results, weakness in the internal rotator and external rotator strength showed an association with recurrent anterior shoulder instability [29]. In our study, a wider area of the external rotation muscle group (ISP/TM) was observed in the chronic group. However, no statistical significance was observed between the two groups.

Other muscles were often able to compensate for weakness in specific muscles. The muscles around the shoulder (pectoralis major, teres major, and latissimus dorsi muscle) may obscure the contributions of the individual rotator cuff muscle during testing of rotation powers using an isokinetic device [22]. Jan et al., who reported on a study on the potential impact of recurrences of post-traumatic anterior dislocation of the shoulder on balance in external and internal rotational muscles, measured the internal and external power in 102 patients after recurrent dislocations using a Cybex Norm dynamometer [30]. Isokinetic testing was performed in a sitting position with the arm held at an angle between 25° and 45° in the plane of the scapula and with the elbow flexed at an angle of 90°. A comparison of dislocated and healthy sides showed that there was no significant difference in the external rotator/internal rotator ratio [30].

Shoulder dislocation can affect both muscle power and neuromuscular control. This is because injuries of the capsuloligamentous complex may cause alteration of the afferent sensory signals to the central nervous system [20]. This type of alteration can be expressed as altered patterns of muscle recruitment, including decreased muscle activity or delayed timing of muscle activation [20]. Therefore, several authors have reported that measurement of the time to peak torque and acceleration time in multiple joints including the shoulder, hip, knee, and ankle joints can be performed using isokinetic devices for assessment of muscular coordination [31]. Lee et al. performed isokinetic performance testing in patients with anterior shoulder instability compared to asymptomatic patients [18]. Twenty male nonathletic patients with traumatic anterior shoulder instability and twenty normal control patients were enrolled. Isokinetic muscle performance testing was performed at an angular velocity of 180°/s with 90° of shoulder abduction. Measurement of muscle strength and neuromuscular control (time to peak torque and acceleration time) of the internal rotators and external rotators was performed. According to their results, decreased neuromuscular control of both internal and external rotations was observed in cases of traumatic dislocation of the shoulder [10]. Therefore, they emphasized the necessity of attempting to restore neuromuscular control during rehabilitation.

Several studies using electromyography (EMG) also reported on the potential for significantly decreased neuromuscular control of the rotator cuff in patients with shoulder instability [32,33]. Brostrom et al., who examined muscle activity in EMG in patients with non-traumatic shoulder instability, reported that the level of activity in the SSC was low, and the activation speed was slow in the 45° abduction position [32]. Fremerey et al., who also analyzed the activity of the shoulder around muscles using EMG, enrolled 43 patients with chronic anterior shoulder instability who underwent capsulolabral reconstruction [34]. A surface EMG device was attached to the patient’s deltoid, supraspinatus, infraspinatus, and biceps brachii muscles. The results of proprioception and assessment of muscle activity showed persistent deficits, including altered EMG patterns, and reduced activity of the deltoid muscle on the operated side, which may explain the challenge in achieving complete restoration of shoulder joint function following surgery [34].

Although only cases of acute and chronic anterior shoulder dislocation were included in this study, Ishikawa et al. focused on the relationship between the direction of shoulder instability and the rotator cuff muscle using MRI [22]. In their study, shoulder dislocation was classified according to direction (anterior, posterior, and multidirectional dislocation) based on three categories. The rotator cuff (SSP, SSC, and ISP + TM) and deltoid (anterior and posterior portions, and total) muscle area were then measured using T1 sagittal and axial MRI. Analysis of the ratios of the SSC to ISP + TM area and the anterior deltoid and posterior deltoid muscles was performed for quantification of the transverse force couple imbalance. According to their results, a smaller ISP + TM was observed for anterior instability, and a smaller SSC was observed for posterior instability; however, no difference in the anterior-to-posterior deltoid area ratio was observed among the three directions. This result was contrary to their hypothesis [22]. They were not able to explain their findings, which could be a cause of further exacerbation of shoulder instability. However, they concluded that shoulder dislocation could cause an imbalanced transverse force couple of the rotator cuff, and that strengthening of the ISP + TM may be helpful in balancing the force couple in patients with anterior instability [22].

The current study has several limitations. First, it was retrospective and included a relatively small sample size. Second, correlations with clinical outcomes, including functional scores and muscle power, were not examined. Third, 3-dimensional reconstruction was not utilized for the measurement of muscle volume, which might have enabled increased precision. However, our study holds significance as the first to examine differences in rotator cuff muscle changes between patients with acute and chronic anterior shoulder instability. The rehabilitation program has focused on improving the patient’s muscle strength and/or imbalance in conservative treatment of shoulder instability. Therefore, despite these limitations, our findings provide valuable insights into this area of research, so that more comprehensive study may be conducted in the future.

## 5. Conclusions

Larger areas of SSP muscle were observed in patients with chronic anterior shoulder instability compared to those with acute instability. In contrast, a smaller area of SSC muscle was observed in the chronic group than in the acute group. Stability and strength are essential components of shoulder function and should be a focus during the postoperative rehabilitation period. Therefore, understanding the pathophysiology of muscle imbalance is important for appropriate targeting of therapeutic interventions.

## Figures and Tables

**Figure 1 diagnostics-14-00648-f001:**
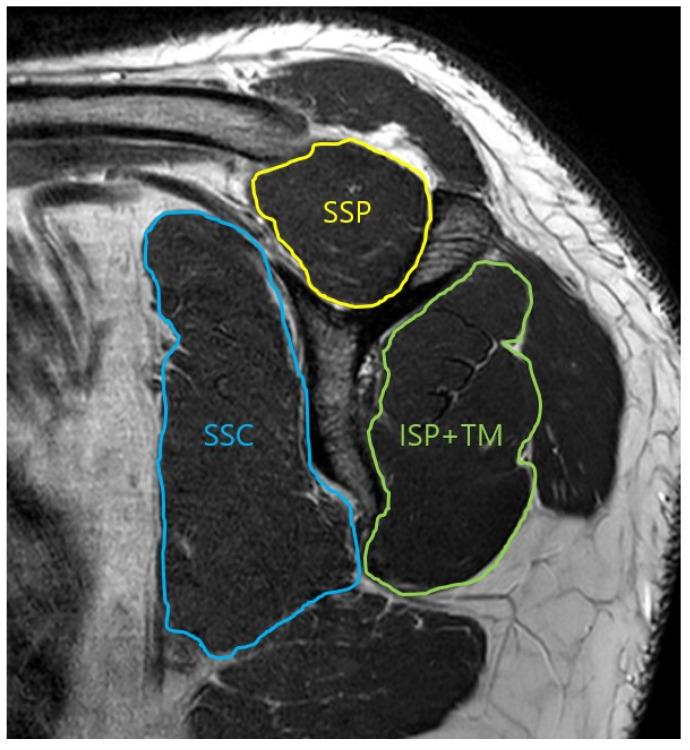
Oblique-sagittal T2 TSE MRI. Measurement of the cross-section area of the rotator cuff muscle in scapular Y view. SSP: supraspinatus, SSC: subscapularis, ISP: infraspinatus, TM: teres minor.

**Figure 2 diagnostics-14-00648-f002:**
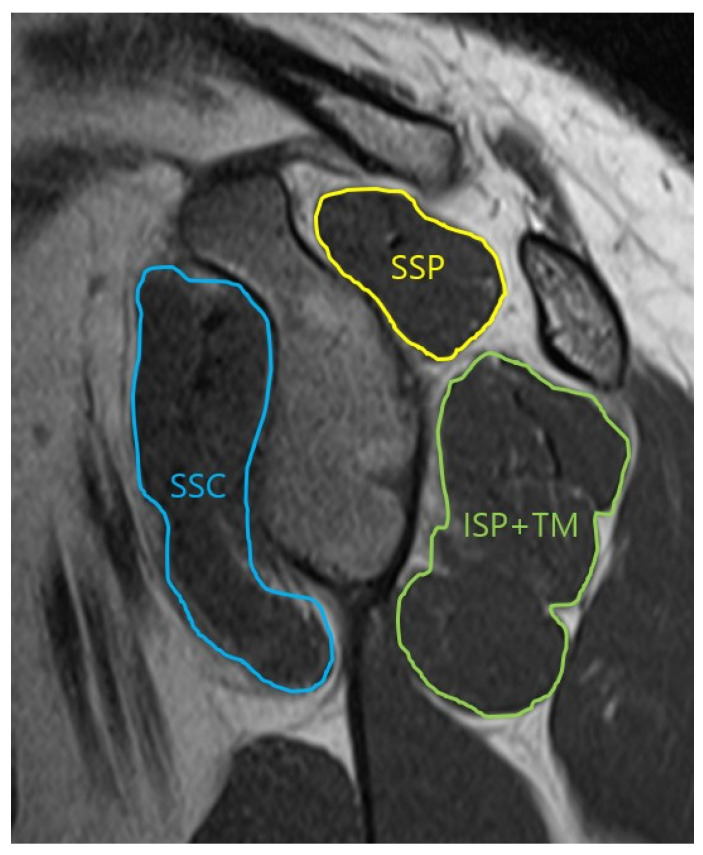
Oblique-sagittal T2 TSE MRI. Measurement of the cross-section area of the rotator cuff muscle in glenoid face view. SSP: supraspinatus, SSC: subscapularis, ISP: infraspinatus, TM: teres minor.

**Table 1 diagnostics-14-00648-t001:** Demographic data.

Parameter	Acute Group (*n* = 28)	Chronic Group (*n* = 28)	*p* Value
Age (year)	25.0 ± 11.4	28.1 ± 8.2	0.496
Sex (*n*)			0.079
Male	22	18	
Female	6	10	
Involved side (*n*)			0.784
Right	18	17	
Left	10	11	
Time between trauma and imaging	8.6 ± 10.9 days	93.3 ± 82.6 mo	0.000 *
No. of dislocation episodes	1.2 ± 0.6	8.8 ± 7.3	0.000 *

* statistically significant.

**Table 2 diagnostics-14-00648-t002:** Inter- and intraclass correlation of the measurement.

Injured Structures	Inter CC	Intra CC	*p* Value
Y view SSP	0.957	0.881	0.0001
Y view SSC	0.929	0.853	0.0001
Y view ISP/TM	0.914	0.869	0.0001
Glenoid face view SSP	0.959	0.911	0.0001
Glenoid face view SSC	0.901	0.899	0.0001
Glenoid face view ISP/TM	0.898	0.904	0.0001

SSP: supraspinatus, SSC: subscapularis, ISP: infraspinatus, TM: teres minor, CC: class correlation.

**Table 3 diagnostics-14-00648-t003:** Rotator cuff muscle volume in the acute and chronic groups.

Rotator Cuff	Acute Group (*n* = 28)	Chronic Group (*n* = 28)	*p* Value
Supraspinatus (%)			
Y view	15.6 ± 2.2	17.2 ± 2.3	0.006 *
Glenoid face view	15.8 ± 2.2	18.5 ± 2.5	0.000 *
Infraspinatus + teres minor			
Y view	34.9 ± 5.3	35.7 ± 3.4	0.497
Glenoid face view	38.7 ± 4.2	39.9 ± 3.8	0.255
Subscapularis			
Y view	49.6 ± 5.3	47.1 ± 3.5	0.044 *
Glenoid face view	45.6 ± 4.4	41.6 ± 3.2	0.000 *

* Statistically significant.

## Data Availability

Data are contained within the article.

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
