# Peer review of "Rotator Cuff Muscle Imbalance in Patients with Chronic Anterior Shoulder Instability"

_diagnostics, 2024, doi:10.3390/diagnostics14060648_

Round 1

Reviewer 1 Report

Comments and Suggestions for Authors

The study investigates the cross-sectional areas of rotator cuff muscles in patients with acute and chronic anterior shoulder instability, aiming to shed light on the potential associations between rotator cuff imbalance and the chronicity of instability. The introduction is well written and sets up a promising study looking into the structure & function relationship between anatomical changes linked to shoulder instability. A major flaw in this study is as pointed out by the authors is the lack of functional data, despite introducing a 'direct measurement' for shoulder function.
There is no link between these structural changes to functional capacity and the discussion itself appeared as a literature review of current papers that incorporate both aspects to address shoulder instability. The study is novel in introducing changes in rotator cuff muscles anatomy via MRI, however, lacks in fundamental data to link these findings to a functional change.

In one particular section in the results an entire paragraph was used for statistics (which showed P<0.000?). This could have been a typo error as no statistics were introduced for that table. Furthermore this information could have been added as a separate column in the table.

Overall the manuscript lacks data & requires Major revisions in both study design & manuscript text (results and discussion sections).  

Please see attached PDF with highlight & comments for reference.

Comments on the Quality of English Language

Introduction was well written for a structure-function paper, however, the study itself reflected on this poorly & lacks data. 

Results and discussion need to be re-written, discussion was more of a review of the current literature failing to draw appropriate links between the anatomical data presented and what is published out there. Discussion segment is also written as if spoken & not appropriate for scientific writing, several instances have been pointed out in attached PDF & requires restructuring. 

Author Response

Thank you for your comments. 

We revised our manuscript according to your comments. 

Please check it. 

Reviewer 2 Report

Comments and Suggestions for Authors

This study evaluated the cross-sectional areas of the rotator cuff muscle in patients with acute and chronic anterior shoulder instability to determine the associations between rotator cuff imbalance and chronicity. However, no previous research has been documented on changes in the rotator cuff according to the chronicity of shoulder anterior instability, it is clinically innovative. But there are still some issues that need to be discussed.

1. Do the exclusion criteria include glenoid labral injuries?

2. Is the definition of acute group accurate? What is the reason for 30 days?

3. How was the side of the Rotator cuff muscle volume measurement selected?

4. What are the relevant parameters such as layer thickness of the 1.5T MRI system (Siemens Magnetom Avanto System; Siemens Medical, Erlangen, Germany)?

5. Does race and BMI affect Rotator cuff muscle volume measurements?

Author Response

This study evaluated the cross-sectional areas of the rotator cuff muscle in patients with acute and chronic anterior shoulder instability to determine the associations between rotator cuff imbalance and chronicity. However, no previous research has been documented on changes in the rotator cuff according to the chronicity of shoulder anterior instability, it is clinically innovative. But there are still some issues that need to be discussed.

  1. Do the exclusion criteria include glenoid labral injuries?

=Thank you for comment. As you know, most of the shoulder dislocations can cause a labral injury. So we did not exclude glenoid labral injury. We only excluded fractures and rotator cuff tears.

  1. Is the definition of acute group accurate? What is the reason for 30 days?

=Thank you for comment. Although it varies depending on the author, acute injury is generally defined as 2 or 3 weeks. However, MRI after injury tends to be delayed several days. So, we defined acute injury as within 30 days. The average time from injury to MRI in our study was about 8 days.

  1. How was the side of the Rotator cuff muscle volume measurement selected?

=Thank you for comment. Using T2 TSE MRI scans, we divided the measurements into SSC, SSP, and ISP+TM. Because the borders of the ISP and TM were often indiscernible, these muscles were combined as a single muscle unit.

  1. What are the relevant parameters such as layer thickness of the 1.5T MRI system (Siemens Magnetom Avanto System; Siemens Medical, Erlangen, Germany)?

=Thank you for comment. We added additional information about MRI scans. Please see Materials and Methods 2.2 (2nd and 3rd sentences)

  1. Does race and BMI affect Rotator cuff muscle volume measurements?

=Thank you for comment. We considered that RC volume might vary depending on each individual. Therefore, we conducted a study on the ratio change of each person’s RC volume (SSP, SSC and TM/ISP). The sum of the three muscles was set to 100%.

Reviewer 3 Report

Comments and Suggestions for Authors

Page 2
Muscle area measurement:
I think is better to creat a separate paragraph regarding MRI protocol (parameter, planes, and time of the scan) and acquisition.
It is important to specify which sequence is used for the measurements.

Fig.1/2: Please report the sequence used.

Results:
There is an imbalance between male and female patients.
Don't you think the greater trophism and volume of the male muscle bellies could affect the results?

Comments on the Quality of English Language

Only minor editing of English language required

Author Response

Page 2
Muscle area measurement:
I think is better to creat a separate paragraph regarding MRI protocol (parameter, planes, and time of the scan) and acquisition.
It is important to specify which sequence is used for the measurements.
=Thank you for comment. We added additional information about MRI scans. Please see Materials and Methods 2.2 (2nd and 3rd sentences)

Fig.1/2: Please report the sequence used.
=Thank you for comment. We added information about MRI in the “figure legend”.

Results:
There is an imbalance between male and female patients.
Don't you think the greater trophism and volume of the male muscle bellies could affect the results?

=Thank you for comment. As per your comment, we considered that RC volume might vary depending on each individual (sex, BMI, height etc.). Therefore, we conducted a study on the ratio change of each person’s RC volume (SSP, SSC and TM/ISP), not RC volume. The sum of the three muscles was set to 100%. Additionally, we believe that changes in the ratio are appropriate for investigating changes in individual RC imbalances.

Round 2

Reviewer 1 Report

Comments and Suggestions for Authors

This reviewer is happy with the manuscript as is, thankyou for revising the original manuscript.